# Association between Negatively Charged Low-Density Lipoprotein L5 and Subclinical Atherosclerosis in Rheumatoid Arthritis Patients

**DOI:** 10.3390/jcm8020177

**Published:** 2019-02-03

**Authors:** Chun-Yu Chang, Chu-Huang Chen, Yi-Ming Chen, Tsu-Yi Hsieh, Ju-Pi Li, Ming-Yi Shen, Joung-Liang Lan, Der-Yuan Chen

**Affiliations:** 1Department of Medical Education and Research, Taichung Veterans General Hospital, Taichung 407, Taiwan; chung6556@gmail.com (C.-Y.C.); blacklark@gmail.com (Y.-M.C.); zuyihsieh@gmail.com (T.-Y.H.); 2Rheumatology and Immunology Center, China Medical University Hospital, Taichung 404, Taiwan; shenmy1124@gmail.com; 3School of Medicine, China Medical University, Taichung 404, Taiwan; 4Lipid Science and Aging Research Center, Kaohsiung Medical University, Kaohsiung 807, Taiwan; cchen@texasheart.org; 5Vascular and Medicinal Research, Texas Heart Institute, Houston, TX 77030, USA; 6Center for Lipid Biosciences, Kaohsiung Medical University Hospital, Kaohsiung 807, Taiwan; 7Faculty of Medicine, National Yang Ming University, Taipei 112, Taiwan; 8Rheumatic Diseases Research Laboratory, Rheumatology and Immunology Center, China Medical University Hospital, Taichung 404, Taiwan; 9Graduate Institute of Clinical Medical Science, China Medical University, Taichung 404, Taiwan; d888203@gmail.com; 10Department of Medical Research, China Medical University Hospital, Taichung 404, Taiwan; 11Translational Medicine Laboratory, Rheumatology and Immunology Center, China Medical University Hospital, Taichung 404, Taiwan

**Keywords:** low-density lipoprotein (LDL), electronegative LDL, L5, subclinical atherosclerosis, rheumatoid arthritis (RA)

## Abstract

L5, the most negatively charged subfraction of low-density lipoprotein (LDL), is implicated in atherogenesis. We examined the relationship between plasma L5 levels and the occurrence of subclinical atherosclerosis in patients with rheumatoid arthritis (RA). Using anion-exchange purification with fast-protein liquid chromatography, we determined the proportion of plasma L5 of LDL (L5%) in 64 RA patients and 12 healthy controls (HC). Plasma L5% and L5 levels were significantly higher in RA patients (median, 1.4% and 1.92 mg/dL) compared with HC (0.9%, *p* < 0.005; and 1.27 mg/dL, *p* < 0.05) and further increased in patients with subclinical atherosclerosis (2.0% and 2.88 mg/dL). L5% and L5 levels decreased in patients after 6-months of therapy (*p* < 0.01). Subclinical atherosclerosis was indicated by plaque and intima-media thickness determined by carotid ultrasonography. Using multivariate analysis, L5% and L5 levels are revealed as the predictors of subclinical atherosclerosis (odds ratio, 4.94 and 1.01; both *p* < 0.05). Receiver operating characteristic curves showed that cut-off values of L5% ≥ 1.45% and L5 levels ≥ 2.58 mg/dL could predict subclinical atherosclerosis in patients (both *p* < 0.001). Immunoblotting showed that the expression levels of lectin-like oxidized LDL receptor-1 (LOX-1) was increased in RA patients. Together, our findings suggest that plasma L5% and L5 levels may be predictors of cardiovascular risk in RA patients.

## 1. Introduction

Atherosclerosis, a chronic inflammatory vascular disease characterized by atheromatous plaque buildup, is associated with an increased risk of cardiovascular events [1]. The subendothelial retention of oxidatively modified low-density lipoprotein (LDL) represents the initial event in atherosclerosis. Rheumatoid arthritis (RA) is an inflammatory articular disease [2,3] that is complicated by accelerated atherosclerosis and an increased risk of cardiovascular disease (CVD) [4,5]. The high CVD prevalence in RA patients would be explained by both conventional CV risk factors and systemic inflammation in this disease [6,7]. Given the lipid paradox hypothesis in RA [8], we recently showed an inverse correlation between RA-related inflammation and LDL cholesterol (LDL-C) levels [9], a finding consistent with those previously reported [6,8]. 

Considering the lipid paradox in RA [8], the possibility remains that LDL-C contains critical atherogenic components that are not reflected in the absolute LDL-C level. Electronegative LDL, which was first separated from its electropositive counterpart by Avogaro and colleagues [10] by using anion-exchange chromatography, has been shown to be atherogenic [11]. Using a modified chromatography assay, others have shown that LDL-C can be divided according to electronegativity into five subfractions, called L1–L5, with L1 being the least electronegative (and most abundant) and L5 being the most electronegative [12,13,14]. The binding of L5 with its specific receptor, lectin-like oxidized LDL receptor-1 (LOX-1), has been shown to induce the release of multiple inflammatory and atherogenic mediators from vascular endothelial cells and monocytes [12,13,14,15]. However, whether electronegative L5 contributes to the increased risk of CVD in RA patients has not been explored.

In addition to being a major receptor for oxidized LDL (ox-LDL) [16], LOX-1 is exclusively responsible for the internalization of L5 [17]. LOX-1 is expressed in vascular endothelial cells, smooth muscle cells, and monocytes in response to atherogenic stimuli [18] and is found primarily in the macrophage-rich lipid core area where monocyte chemotactic protein-1 expression and apoptotic events are prominent [19]. The increased expression of LOX-1 observed in atherosclerotic plaque [20,21] and the reduced number of atherosclerotic lesions observed in LOX-1 knockout mice [22] suggested a critical role for LOX-1 in atherosclerosis.

Ultrasonography of the carotid artery indicates the presence of subclinical atherosclerosis [23,24,25]. Common carotid artery intima-media thickness (ccIMT) measurements have also been shown to indicate the extent of coronary atherosclerosis [24,25]. In RA patients, increased ccIMT together with the evidence of plaque have been shown to be predictors of the emergence of CVD [24,25]. Thus, increased ccIMT and the presence of carotid plaque may be used as the gold standard for identifying subclinical atherosclerosis and patients at high CVD risk based on current guidelines [23,24].

In this pilot study, we examined the relationship between plasma L5 levels and the occurrence of subclinical atherosclerosis in RA patients and whether L5 may contribute to the increased CVD risk. We compared the differences in the proportion of plasma L5 in total LDL-C (L5%) and L5 levels between RA patients and healthy subjects and between RA patients with and without subclinical atherosclerosis. We also examined the association of plasma L5 levels with the extent of subclinical atherosclerosis and 10-year CVD risk and determined potential predictive factors and their optimal cut-off values for predicting subclinical atherosclerosis in RA patients.

## 2. Materials and Methods

### 2.1. Study Population.

Sixty-four patients who met the 2010 revised criteria of the American College of Rheumatology for RA [26] were consecutively enrolled. Each patient had previously received corticosteroids, nonsteroidal anti-inflammatory drugs, and at least one conventional synthetic disease-modifying anti-rheumatic drug (csDMARD). Patients with a recent history (i.e., within one year before enrollment) of coronary heart disease or ischemic stroke were excluded. Disease activity was assessed by using the 28-joint disease activity score (DAS28) [27]. Follow-up for the emergence of CVD, which included acute myocardial infarction and ischemic stroke, was done for at least two years. Twelve sex- and age-matched healthy volunteers who had no rheumatic disease were enrolled as healthy controls. This study was approved by the Institutional Review Board of Taichung Veterans General Hospital (CE14237A), and written consent was obtained from each participant according to the Declaration of Helsinki.

### 2.2. Determination of Plasma Lipid Profiles and Atherogenic index (AI)

All blood samples were collected from patients in the early morning after an overnight fast for 12 h. Plasma levels of total cholesterol, triglyceride, high-density lipoprotein cholesterol (HDL-C), and LDL-C were measured by using enzymatic methods with a chemistry analyzer (Hitachi 7600, Hitachi, Tokyo, Japan) according to the manufacturer’s instructions. The AI (i.e., the ratio of total cholesterol/HDL-C) was calculated.

### 2.3. Measurement of 10-Year Risk of CVD (QRISK-2 Score)

Global 10-year risk for a heart attack or stroke was estimated by calculating the QRISK-2 scores [28] on the website: https://www.qrisk.org [29] and by calculating the Framingham score [30] on the website: https://reference.medscape.com/calculator/framingham-cardiovascular-disease-risk [31]. Briefly, factors including age, sex, ethnicity, physical characteristics, total cholesterol/HDL-C ratio, self-reported smoking status, diabetic status, the presence of kidney disease, and family history of heart disease were considered in determining the QRISK-2 score of each study participant.

### 2.4. Isolation and Fractionation of LDL-C

Plasma was obtained from freshly collected whole blood samples, and 1% antibiotics (penicillin/streptomycin stock solution, Gibco #15140), 0.5 mM EDTA and nitrogen were added to avoid ex vivo oxidation and degradation. LDL-C particles were isolated by using sequential potassium bromide density ultracentrifugation. Purified LDL-C was dialyzed against a degassed solution of 20 mM Tris-HCl and 0.5 mM EDTA at 4 °C with six buffer changes during a 72-h period.

### 2.5. Anion-Exchange Chromatography Purification of LDL-C Subfractions

LDL-C subfractions were separated by using UnoQ12 anion-exchange columns (Bio-Rad Laboratories, Inc., Hercules, CA, USA) with an NGC Quest 10 chromatography system (Bio-Rad). The columns were pre-equilibrated with buffer A (0.02 M Tris-HCl (pH 8.0) and 0.5 mM EDTA) in a 4 °C cold room. After dialysis with buffer A, 100 mg of LDL in 10 mL was injected onto a UnoQ12 column and eluted at a flow rate of 2 mL/min with a multistep gradient of buffer B (1 M NaCl in buffer A). Five LDL-C fractions were eluted with a multistep gradient of buffer B according to electronegativity. L1 was the effluent collected between fractions 11 to 14 (18–28 min); L2, fractions 15 to 16 (28–32 min); L3, fractions 17 to 24 (32–48 min); L4, fractions 25 to 30 (48–60 min); and L5, fractions 31 to 40 (60–80 min). Protein concentrations were determined by using the Bradford method [32]. The respective fractions were then concentrated with Centriprep filters (YM-30, Merck Millipore Corp., Burlington, MA, USA) and sterilized by passage through 0.22 μm filters.

### 2.6. Agarose Gel Electrophoresis of LDL-C Subfractions

To confirm different electric charges among the separated LDL-C subfractions, 1 mg of each was loaded onto a 0.72% agarose gel (90 mM Tris, 80 mM borate, and 2 mM EDTA (pH 8.2)). Bovine serum albumin (BSA), which is negatively charged, was used as a reference. Gel electrophoresis was performed at 100 V for 2 h, followed by staining with Simply Blue SafeStain (Invitrogen Corp., Auckland, NY, USA).

### 2.7. Determination of Protein Expression with Immunoblotting

Total protein was extracted from peripheral blood mononuclear cells (PBMCs). The proteins were separated with 10% SDS-PAGE and were then transferred to PVDF membranes (Bio-Rad). Immunoblotting was performed by using primary antibodies against LDLR (anti-LDLR; 1:1000, Abcam, Cambridge, MA, USA), anti-LOX-1 (1:1000, Abcam), and anti-GAPDH (1:5000, Santa Cruz Biotechnology, Dallas, TX, USA) and HRP-conjugated secondary antibodies against mouse or rabbit IgG (1:10000, Santa Cruz). Luminescence was detected by using the Fujifilm LAS-3000 image detection system, and image processing and data quantification were performed by using Multi Gauge v.2.02 software (FUJiFILM, Tokyo, Japan). LDLR expression levels were normalized to those of GAPDH (as a loading control), and LOX-1 expression levels were normalized to those of its proform.

### 2.8. Cell Culture

Human monocytic THP-1 cells (ATCC TIB-202; American Type Culture Collection, Rockville, MD, USA) were grown in RPMI 1640 (Gibco, ThermoFisher Scientific Inc., Pittsburgh, PA, USA) supplemented with 10% fetal bovine serum, 1% penicillin/streptomycin antibiotics in an incubator containing 5% CO2 at 37 °C. To readily induce differentiation into macrophages, THP-1 cells (1 × 106 cells/mL) were grown in media and treated with 10 ng/mL phorbol myristate acetate (Millipore Sigma-Aldrich, St. Louis, MO, USA) overnight.

### 2.9. LOX-1 and LDLR Protein Expression in THP-1 Cells Treated with L1 or L5

THP-1 cells at a density of 1 × 10^7^ cells/mL were treated with native human plasma L1 or L5 (50 μg/mL) at 37 °C overnight, and protein extracts were prepared. Protein levels of LOX-1 and LDLR were determined by using immunoblotting as described above.

### 2.10. Ultrasound Vascular Imaging of Carotid Arteries

Ultrasound vascular imaging of the carotid arteries included the measurement of ccIMT and the detection of focal plaque in the extracranial carotid tree. Carotid plaque was defined as a localized thickening >1.5 mm that did not uniformly involve the whole artery. A ccIMT >0.9 mm or the presence of carotid plaque are used as the gold standard for defining subclinical atherosclerosis [23,24].

### 2.11. Statistical Analysis

The nonparametric Mann-Whitney U test was used for 2-group comparisons of plasma L5%, L5 levels, and LOX-1 and LDLR expression. For evaluating the changes of plasma L5% and L5 levels during the follow-up period, the Wilcoxon signed rank test was employed. The nonparametric Spearman’s rank correlation test was used to determine the correlation of plasma L5% or L5 levels with the expression of LOX-1 or LDLR and with the extent of carotid atherosclerosis or 10-year CVD risk scores (QRISK-2 scores and Framingham score). We also constructed a univariate logistic regression model to evaluate factors contributing to the presence of subclinical atherosclerosis detected by ultrasonography, including traditional risk factors such as age, RA disease duration, plasma L5% and L5 levels, and RA-related inflammatory parameter. Age, RA disease duration and the statistically significant variables in the univariate analysis were entered into the multivariate regression analysis. The optimal cut-off values of plasma L5% and L5 levels for predicting the presence of subclinical atherosclerosis in RA patients were determined by using receiver operating characteristic (ROC) curve analysis. The diagnostic sensitivity and specificity were determined by MedCalc statistical software version 9.3 (MedCalc Software, Ostend, Belgium). A *p*-value <0.05 was considered statistically significant.

## 3. Results

### 3.1. Clinical Characteristics of RA Patients

Of the 64 RA patients, 43 (67.2%) were positive for rheumatoid factor (RF), and 40 (62.5%) were positive for anti-citrullinated peptide antibodies (ACPA). According to the results of carotid ultrasonography, 30 (46.9%) patients had subclinical atherosclerosis, of whom 3 (10.0%) had CVD (two with acute myocardial infarction and one with ischemic stroke) during the 2-year follow-up period. CVD did not develop in any of 34 patients without subclinical atherosclerosis (Table 1). No significant differences were observed in demographic data, clinical characteristics, or comorbidities among these three groups, with the exception of an older average age and a higher proportion of hypertension in RA patients with subclinical atherosclerosis (all *p* < 0.05).

### 3.2. Comparison of Lipid Profiles, QRISK-2 Scores, and AI among RA Patients with or without Subclinical Atherosclerosis and Healthy Controls

In Table 1, RA patients with subclinical atherosclerosis had significantly lower HDL-C levels than RA patients without subclinical atherosclerosis or healthy controls. RA patients also had significantly higher QRISK-2 scores (median 7.2, interquartile range (IQR) 3.7–10.4) than did healthy controls (median 3.8, IQR 2.9–5.0, *p* < 0.01) and Framingham scores (median 8.9, IQR 4.8–14.6) compared with healthy controls (median 3.8, IQR 2.9–5.0, *p* < 0.01 and 3.6, IQR 2.5–5.0, *p* < 0.001; respectively). In addition, QRISK-2 scores and Framingham scores were even higher in patients with subclinical atherosclerosis than in those without (*p* < 0.005 and *p* < 0.001, respectively). However, no significant differences were observed in AI or in plasma levels of total cholesterol, triglyceride, or LDL-C between RA patients and healthy controls or between RA patients with subclinical atherosclerosis and those without.

### 3.3. Increased Plasma L5% and L5 Levels in RA Patients

Representative distribution (Figure 1A,B) and electrophoretic mobility patterns (Figure 1C) are shown for LDL-C subfractions L1 and L5 from plasma of RA patients and healthy controls. Plasma L5% and L5 levels were significantly higher in RA patients (L5%: Median 1.4%, IQR 0.8–2.2%; L5: 1.92 mg/dL, IQR 1.16–3.13 mg/dL) than in healthy controls (L5%: Median 0.9%, IQR 0.6–1.1%, *p* < 0.005; L5: 1.27 mg/dL, IQR 0.80–1.51 mg/dL, *p* < 0.05; Figure 1D–E). Furthermore, L5% and L5 levels were significantly higher in RA patients with subclinical atherosclerosis (L5%: Median 2.0%, IQR 1.3–4.5%; L5: 2.88 mg/dL, IQR 1.73–5.67 mg/dL; *p* < 0.001) than in RA patients without subclinical atherosclerosis (L5%: median 0.9%, IQR 0.7–1.7%; L5: 1.33 mg/dL, IQR 0.92–2.25 mg/dL; *p* < 0.001).

### 3.4. The Change of Plasma L5% and L5 Levels in RA Patients after 6-Month Therapy

Fourteen patients were available for examining plasma L5% and L5 levels before (at baseline) and after 6-month therapy. Six patients received tumor necrosis factor-α inhibitor (adalimumab) at a dose of 40 mg every other week, and 8 received interleukin-6 receptor inhibitor (tocilizumab) at a dose of 4 mg/kg once monthly during the first 3 months and then 8 mg/kg once monthly afterward, all with concomitant MTX at a stable dose of 7.5–15 mg weekly. The dosage of csDMARDs as well as oral corticosteroids remained unchanged throughout the period of investigation. Our results showed that plasma L5% and L5 levels significantly decreased (median 2.2%, IQR 1.4–4.1% vs. 1.2%, IQR 0.7–2.4% and 3.53 mg/dL, IQR 2.02–4.96 mg/dL vs. 1.65 mg/dL, IQR 1.10–3.63 mg/dL; respectively, both *p* < 0.01) in RA patients with a decrease of disease activity after 6-months of therapy.

### 3.5. Increased Expression of LOX-1 in the PBMCs from RA Patients

Representative immunoblots in Figure 2A show the expression of LDLR and the L5 receptor LOX-1 in PBMCs of RA patients and healthy controls. No significant difference was observed in LDLR expression between RA patients and healthy controls (Figure 2B). However, LOX-1expression in PBMCs was 2~3 times greater in RA patients (median 2.29, IQR 0.63–4.09) than in healthy controls (median 0.61, IQR 0.36–1.28, *p* < 0.05; Figure 2C).

To determine whether LOX-1 expression can be induced specifically by L5 from RA patients (and not by L1), we examined LOX-1 expression in the lysates of THP-1 cells treated with plasma L5 or L1 from RA patients. Our results showed a higher expression of LOX-1 in the lysates of THP-1 cells treated with L5 (median 0.85, IQR 0.49–2.28) than in the lysates of PBS-treated THP-1 cells (median 0.55, IQR 0.35–0.94, *p* = 0.059). No enhancement of LOX-1 expression was also observed in THP-1 cells treated with L1 (median 0.42, IQR 0.35–0.51, Figure 2D–E).

### 3.6. Correlation of Plasma L5% and L5 Levels with the Extent of Atherosclerosis, LOX-1 Expression or RA Disease Activity

As shown in Table 2, we observed positive correlations of plasma L5% and L5 levels with the extent of carotid atherosclerosis (ccIMT) and 10-year CVD risk (QRISK-2) scores, but not with Framingham score. We revealed a positive correlation of plasma L5% and L5 levels with LOX-1expression but not LDLR expression. In addition, there was a positive correlation between RA activity score (DAS28) and plasma L5% and L5 levels. However, LDL-C levels were not significantly correlated with the extent of carotid atherosclerosis, 10-year CVD risk scores, or DAS28 in RA patients.

### 3.7. Multivariate Logistic Regression and ROC Curve Analyses

By using a multivariate logistic regression model with the inclusion of age and RA duration, we identified age, L5%, and L5 levels as significant predictors of subclinical atherosclerosis (Table 3). When we performed ROC curve analysis, the optimal cut-off values for predicting the emergence of subclinical atherosclerosis were 1.45% for L5% (area under the curve (AUC) 0.776, sensitivity 73.3%, and specificity 73.5%, *p* < 0.001) and 2.58 mg/dL for L5 levels (AUC 0.788, sensitivity 56.7%, and specificity 88.2%, *p* < 0.001) (Figure 3A,B)

## 4. Discussion

Although a link between CVD and RA has been well established, the pathogenic association between dyslipidemia and atherosclerosis in RA has been relatively unexplored. In this study, we have shown for the first time that plasma levels of L5, the most electronegative subfraction of LDL-C, were significantly higher in RA patients than in healthy controls. In addition, plasma L5% and L5 levels but not LDL-C levels were significantly higher in RA patients with subclinical atherosclerosis than in those without. Furthermore, plasma L5% and L5 levels were positively correlated with the extent of carotid artery atherosclerosis (as indicated by the ccIMT), disease activity score (DAS28), and the 10-year CVD risk (QRISK-2) scores in RA patients. Using multivariate regression analysis, we also revealed that plasma L5% and L5 levels are potential predictors of subclinical atherosclerosis in RA patients. After 6-months of therapy in RA patients, plasma L5% and L5 levels significantly decreased. Thus, our observations suggest that the L5 subfraction of LDL-C is related to atherosclerosis in RA patients.

Dyslipidemia has been well established as a traditional risk factor for atherosclerosis in RA [33]. The subfraction of LDL-C that carries the greatest negative charge may also contribute to atherogenicity in dyslipidemia [11,14]. Likewise, we have shown that L5% and L5 levels are elevated in RA patients, who are at increased risks of atherosclerosis [3,4]. Moreover, we have shown that plasma L5% and L5 levels are increased to an even greater extent (median 2.0% and 2.88 mg/dL, respectively) in RA patients with subclinical atherosclerosis. These findings are in accordance with those previously showing that plasma L5 levels are significantly increased in uremia patients with considerable CVD risks [34]. Our results presented here further support that circulating L5 is proatherogenic [35] and that it is the predominant subfraction of LDL-C capable of inducing endothelial dysfunction and atherogenic responses in cultured vascular cells [12,13,18,19]. Previously, L5 levels were similarly shown to be elevated in asymptomatic patients with type 2 diabetes mellitus [13,36], in whom CVD risk was equivalent to that in patients with RA [37]. Therefore, an increased L5 level is not specific to RA patients.

To identify potential predictors of the occurrence of subclinical atherosclerosis in RA patients, we performed multivariate regression analysis. The data revealed that age is a significant predictor of subclinical atherosclerosis in RA patients, which has been shown previously [38]. In addition, we firstly revealed that L5% and L5 levels are significant predictors of subclinical atherosclerosis in RA patients (odds ratio 4.94 and 1.01, respectively), similar to the finding that L5% significantly increased the adjusted hazard ratio of ischemic peripheral arterial disease [39]. Furthermore, the optimal cut-off value that we determined for predicting subclinical atherosclerosis may have clinical implications. Using ROC analysis, we found that RA patients with plasma L5% above 1.45% and L5 levels above 2.58 mg/dL may have a high probability of subclinical atherosclerosis, with moderate sensitivity and specificity. However, these results are preliminary, and our findings should be substantiated by further confirming their external validity.

We used Framingham risk score and QRISK-2 score to estimate the 10-year CVD risk and found that both scores were significantly higher in RA patients with subclinical atherosclerosis than in those without. The QRISK-2 scores were also positively correlated with plasma L5% or L5 levels, whereas Framingham score were not. This discrepancy may be explained by the more inclusive calculation design of QRISK-2 scoring, additionally enrolling RA disease, ethnicity, chronic kidney disease, and atrial fibrillation as factors in the risk assessment [28,40]. During the follow-up period of more 2 years, CVD developed in three RA patients, all who had carotid atherosclerosis and markedly higher L5% (22.1%, 9.8%, and 7.5%) and L5 levels (21.9 mg/dL, 10.8 mg/dL, and 12.5 mg/dL) than did patients who did not have a CVD event.

In addition to elevated L5% and L5 levels, increased LOX-1 expression in PBMCs was also observed in RA patients compared with healthy controls, a finding similar to that previously reported [41]. Plasma L5 from RA patients but not plasma L1 enhanced LOX-1 expression by two-fold, which supports earlier findings indicating that L5 is internalized exclusively through LOX-1 [13,18]. Moreover, we found that LOX-1 expression levels in patients were positively correlated with plasma L5% or L5 levels and were significantly associated with the 10-year CVD risk (QRISK-2) scores. As reported previously, LOX-1 expression is inducible by the binding of L5, followed by the internalization of the LOX-1–L5 complex and the initialization of atherogenesis [13,18,42]. These observations suggest the probable involvement of LOX-1 in the subclinical atherosclerosis in RA patients.

Despite the novel findings presented here, some limitations should be considered. First, the sample size of patients in whom we observed the emergence of CVD was small, and the duration of follow-up was relatively short. Therefore, we could not assess the actual risks of CVD in our RA patients. Although no significant differences were observed in the proportion of csDMARDs or in the dose of corticosteroids between patients with subclinical atherosclerosis and those without, the probable effects of csDMARDs or corticosteroids on plasma L5 levels remain to be explored. Similar to previous studies of the effects of statins on plasma L5% or L5 levels [43], we did not evaluate the changes in plasma L5 levels after the use of statins. To confirm our findings, a long-term study of a larger group of RA patients is required. Finally, given that this study was cross-sectional in design and the increase of L5 level might reflect a chronic inflammation state, the exact causal association of L5 with accelerated atherosclerosis in RA patients warrants further investigation.

## 5. Conclusions

Our findings suggest that plasma L5% and L5 levels may be independent risk factors of subclinical atherosclerosis in RA patients. Given that plasma L5 from RA patients enhanced LOX-1 expression, L5 may contribute to the atherogenic process in this disease. Long-term follow-up is needed to determine the predictive value of plasma L5% or L5 levels for CVD risk in RA patients.

## Figures and Tables

**Figure 1 jcm-08-00177-f001:**
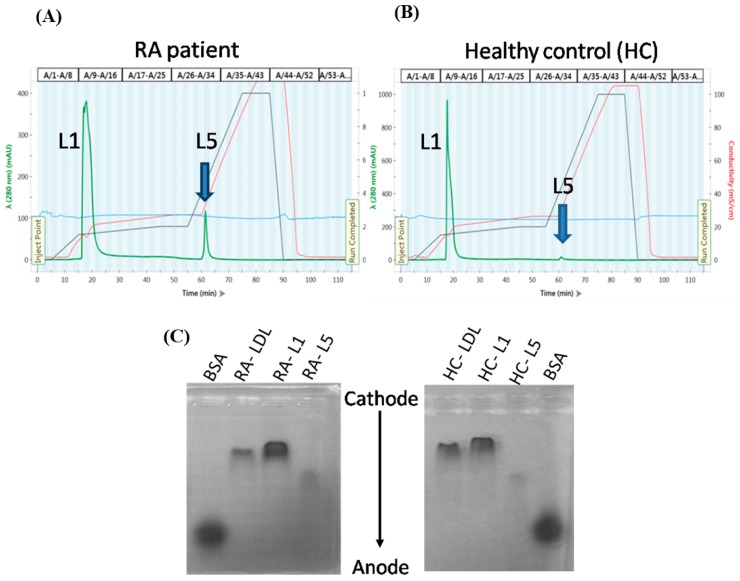
Analysis of LDL subfractions L1 and L5 from RA patients and HC individuals. LDL subfractions L1 and L5 were eluted at the indicated time points according to electronegativity by using anion-exchange fast-protein liquid chromatography. Chromatograms are shown for a (**A**) RA patient and (**B**) healthy control individual. (**C**) LDL subfractions were subjected to agarose gel electrophoresis at 100 V for 2 h (BSA was used as a reference). Comparisons of plasma L5% (**D**) and L5 levels (**E**) between RA patients and healthy controls are shown. The data are presented as box-plot diagrams, in which the box encompasses the 25th percentile (lower bar) to the 75th percentile (upper bar). The horizontal line within the box indicates the median value for each group. BSA: Bovine serum albumin; HC: Healthy control; LDL: Low-density lipoprotein; RA: Rheumatoid arthritis. HC: Healthy control; RA: Rheumatoid arthritis. * *p* < 0.05 and ** *p* < 0.005 vs. HC, determined by using the nonparametric Mann-Whitney U test.

**Figure 2 jcm-08-00177-f002:**
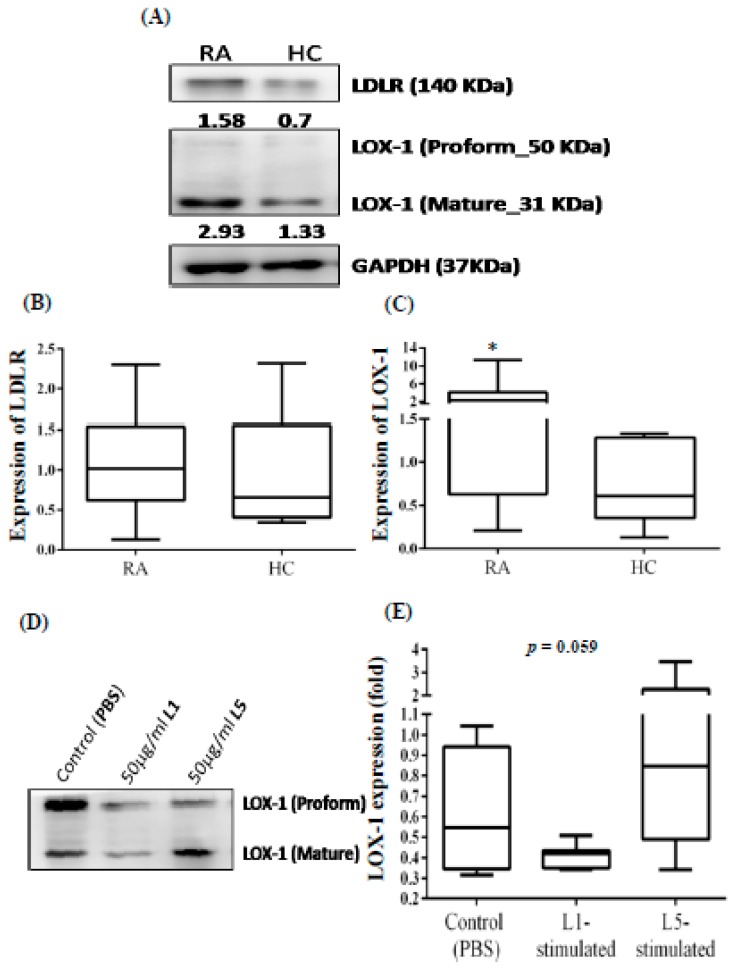
Increased expression of LOX-1 in PBMCs of RA patients compared with HC individuals. Representative immunoblots showed (**A**) the expression of LDLR and the receptor for L5 (LOX-1) in the PBMCs of RA patients and healthy control individuals. Quantitative comparisons are shown for the expression of (**B**) LDLR and (**C**) LOX-1 in PBMCs between RA patients and HC individuals. (**D**) Representative immunoblots showing LOX-1 expression in the lysates of THP-1 cells treated with plasma L5 or L1. Quantitative comparison is shown for (**E**) LOX-1 expression in THP-1 cells treated with control (PBS), L5, or L1. The data are presented as box-plot diagrams, in which the box encompasses the 25th percentile (lower bar) to the 75th percentile (upper bar). The horizontal line within the box indicates the median value for each group. HC: Healthy control; LDLR: Low-density lipoprotein receptor; LOX-1: Lectin-like oxidized low-density lipoprotein receptor 1; PBMCs: Peripheral blood mononuclear cells; PBS: Phosphate-buffered saline; RA: Rheumatoid arthritis. * *p* < 0.05 vs. healthy control individuals, determined by using the nonparametric Mann-Whitney U test.

**Figure 3 jcm-08-00177-f003:**
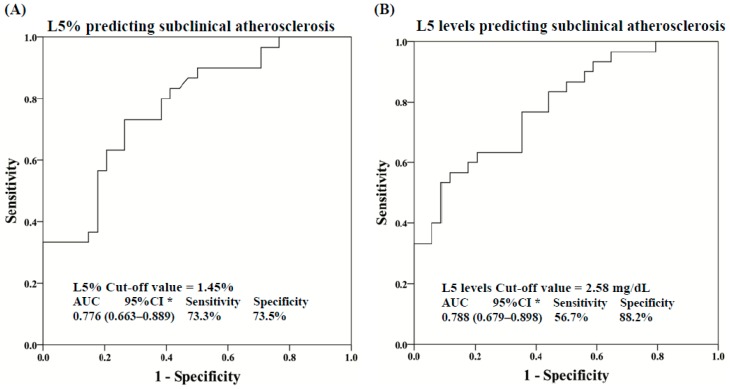
ROC curve analysis for plasma L5% and L5 levels in RA patients. ROC curve analysis of the cut-off values of plasma (**A**) L5% and (**B**) L5 levels for predicting the emergence of subclinical atherosclerosis in RA patients. AUC: Area under the ROC curve; CI: Confidence interval; RA: Rheumatoid arthritis; ROC: Receiver operating characteristic. *p*-value was determined by using the x^2^ test with Yate’s correction of contingency.

**Table 1 jcm-08-00177-t001:** Demographic data and laboratory data in rheumatoid arthritis (RA) patients with or without subclinical atherosclerosis as shown by carotid ultrasonography ^a^.

	RA with Subclinical Atherosclerosis (*n* = 30)	RA without Subclinical Atherosclerosis (*n* = 34)	Healthy Controls (*n* = 12)
Median age at entry, years	63.5 ± 9.1 **^b^**	56.6 ± 11.5	55.1 ± 6.5
Women	23 (76.7%)	27 (79.4%)	9 (75.0%)
Duration of RA, months	81.8 ± 20.5	72.3±22.6	NA
Body mass index, kg/m^2^	23.0 ± 2.8	21.7 ± 3.0	21.4 ± 2.3
RF positivity	21 (70.0%)	22 (64.7%)	NA
ACPA positivity	19 (63.3%)	21 (61.8%)	NA
ESR at entry, mm/1st hour	18.0 ± 12.6	18.0 ± 20.2	NA
CRP at entry, mg/dL	0.52 ± 0.53	0.42 ± 0.67	NA
DAS28 at entry	3.50 ± 0.89	3.52 ± 1.29	NA
Daily steroid dose, mg/day	3.8 ± 1.6	4.0 ± 2.3	NA
csDMARDs used at entry			NA
Methotrexate	25 (83.3%)	28 (82.4%)	NA
Sulfasalazine	14 (46.7%)	15 (44.1%)	NA
Hydroxychloroquine	12 (40.0%)	13 (38.2%)	NA
Biologics used at entry			
TNF-α inhibitors	10 (33.3%)	12 (35.3%)	NA
IL-6 receptor inhibitor	8 (26.7%)	10 (29.4%)	NA
Rituximab	2 (6.7%)	2 (5.9%)	NA
Hypertension	15 (50.0%) ^†^	7 (20.6%)	1 (8.3%)
Diabetes mellitus	5 (16.7%)	2 (5.9%)	0 (0.0%)
Current smoker	4 (13.3%)	4 (11.8%)	1 (8.3%)
Total cholesterol, mg/dL	219 (193–245)	225 (201–247)	208 (201–231)
HDL-C, mg/dL	58.5 (48.0–66.0)	72.5 (56.8–86.0) **^c^**	59.0 (46.3–77.8)
Triglyceride, mg/dL	123 (87–170)	94 (67.5–145)	90 (72.8–126)
LDL-C, mg/dL	142 (111–168)	148 (106–154)	131 (120–155)
Atherogenic index	3.8 (2.8–4.7)	3.1 (2.6–3.9)	3.5 (2.9–4.8)
QRISK-2 scores	9.3 (5.5–14.0) **^d^**	5.4 (2.3–8.7)	3.8 (2.9–5.0)
Framingham score (%)	13.4 (8.5–19.4) **^e^**	5.4 (3.8–5.4)	3.6 (2.5–5.0)
CVD events	3 (10.0%) **^f^**	0 (0.0%)	0 (0.0%)

The presence of subclinical atherosclerosis was determined by using carotid ultrasonography. ^a^ Data are presented as the median (interquartile range, IQR), mean ± SD or number (percentage). ^b^
*p* < 0.05, ^c^
*p* < 0.05, vs. RA patients with subclinical atherosclerosis or healthy controls, as determined by using the Mann-Whitney U test. ^d^
*p* < 0.005, ^e^
*p* < 0.001, vs. RA patients without subclinical atherosclerosis or healthy controls. ^f^ Included two patients with acute myocardial infarction and one with ischemic stroke. ACPA: Anti-citrullinated peptide antibodies; CRP: C-reactive protein; csDMARDs: Conventional synthetic disease-modifying anti-rheumatic drugs; TNF-α: tumor necrosis factor-α; IL-6: interleukin-6; CVD: Cerebrovascular or cardiovascular disease; DAS28: Disease activity score for 28-joints; ESR: Erythrocyte sedimentation rate, HDL-C: High-density lipoprotein cholesterol; LDL-C: Low-density lipoprotein cholesterol; RF: Rheumatoid factor.

**Table 2 jcm-08-00177-t002:** Correlation of plasma levels of LDL-C, L5, L5%, or LOX-1 expression with the extent of subclinical atherosclerosis and 10-year CVD risk (QRISK-2) scores in RA patients (*n* = 64).

	Plasma LDL-C Levels	LDLR Expression	Plasma L5%	Plasma L5 Levels	LOX-1 Expression
Right ccIMT, mm	0.217	0.150	0.537 ^b^	0.611 ^b^	0.457 ^b^
Left ccIMT, mm	0.245	0.223	0.457 ^b^	0.540 ^b^	0.507 ^b^
QRISK-2 scores	0.072	0.244	0.256 ^a^	0.278 ^a^	0.339 ^a^
Framingham score (%)	0.160	0.336^a^	0.107	0.188	0.251
Atherogenic index	0.635 ^b^	0.294 ^a^	−0.024	0.281 ^a^	0.318 ^a^
LDL-C levels	-	0.181	−0.107	0.298 ^a^	0.117
LDLR expression	0.181	-	0.102	0.182	0.298 ^a^
Plasma L5%	−0.107	0.102	-	0.895^b^	0.497 ^b^
Plasma L5 levels	0.298 ^a^	0.182	0.895 ^b^	-	0.588 ^b^
LOX-1 expression	0.117	0.298 ^a^	0.497 ^b^	0.588 ^b^	-
Body mass index	0.056	0.108	0.300 ^a^	0.317 ^a^	0.458 ^b^
DAS28 at entry	−0.225	−0.015	0.361 ^b^	0.254 ^a^	0.166

^a^*p* < 0.05 and ^b^
*p* < 0.01, determined by using the nonparametric Spearman’s correlation test. Atherogenic index corresponds to the ratio of total cholesterol/high-density lipoprotein cholesterol. RA: Rheumatoid arthritis; ccIMT: Common carotid artery intima-media thickness; LDLR: LDL-C receptor; LOX-1: Lectin-like oxidized low-density lipoprotein recptor-1; DAS28: Disease activity score for 28-joints.

**Table 3 jcm-08-00177-t003:** Logistic association of traditional cardiovascular risk factors, RA-related factors, plasma L5% and L5 levels with the presence of subclinical atherosclerosis in RA patients.

Risk Factors	Odds Ratio	95% Confidence Interval	*P* Value
Univariate			
Age	1.07	1.01–1.14	0.017
Sex (female)	0.26	0.08–0.86	0.027
Body mass index	1.17	0.98–1.40	0.085
Smoking	1.15	0.26–5.08	0.850
Hypertension	3.86	1.29–11.55	0.016
Diabetes mellitus	3.20	0.57–17.89	0.185
RA duration	1.02	0.997–1.05	0.089
Steroid daily dose	0.94	0.74–1.21	0.656
RF positivity	1.27	0.44–3.64	0.653
ACPA positivity	1.07	0.39–2.95	0.897
ESR	1.00	0.97–1.03	0.994
CRP	1.30	0.58–3.03	0.505
DAS28 score	0.99	0.63–1.54	0.964
Total cholesterol	1.00	0.99–1.01	0.622
Total triglyceride	1.01	0.996–1.01	0.285
HDL-C	0.97	0.94–0.995	0.022
LDL-C	1.00	0.99–1.00	0.579
L5%	2.95	1.34–6.50	0.007
L5 levels	1.01	1.002–1.01	0.007
QRISK-2 scores	1.16	1.05–1.29	0.005
Atherogenic index	1.60	0.96–2.65	0.069
Multivariate			
Age	1.27	1.04–1.55	0.021
Sex (female)	0.15	0.02–1.09	0.061
RA duration	1.01	0.97–1.05	0.580
Hypertension	5.21	0.84–32.32	0.076
HDL-C	0.96	0.91–1.01	0.120
L5%	4.94	1.48–16.48	0.009
L5 levels	1.01	1.001–1.02	0.010
QRISK-2 scores	0.96	0.92–1.01	0.120

Atherogenic index corresponds to the ratio of total cholesterol/high-density lipoprotein cholesterol (HDL-C). RA: Rheumatoid arthritis; ACPA: Anti-citrullinated peptide antibodies; CRP: C-reactive protein; DAS28: Disease activity score for 28-joints; ESR: Erythrocyte sedimentation rate; LDL-C: Low-density lipoprotein cholesterol; RF: Rheumatoid factor.

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
