# Peer review of "Association between Negatively Charged Low-Density Lipoprotein L5 and Subclinical Atherosclerosis in Rheumatoid Arthritis Patients"

_jcm, 2019, doi:10.3390/jcm8020177_

Round 1
Reviewer 1 Report
This is an interesting clinical observation. The MS is well written and informative for physicians. I have some concerns before consideration of publication.
1. Looking at Table 1, RA patients with subclinical atherosclerosis was significantly older than patients without subclinical atherosclerosis or healthy controls. I wonder age of patients or duration of RA might be attributable to the appearance of subclinical atherosclerosis. Alternatively, age of patients or duration of RA might be influence serum level of low-density lipoprotein L5. This issue should be clarified.
2. The authors observed a decrease of serum level of low-density lipoprotein L5 after anti-RA treatment in patients with RA. Detailed treatment procedure in these patients should be represented. I wonder serum level of low-density lipoprotein L5 might reflect only chronic inflammation state, not some pathogenesis of RA. This issue should be clarified. Also, whether some association between serum level of low-density lipoprotein L5 and serum level of C-reactive protein, erythrocyte sedimentation rate, or DAS28 existed. This issue should be clarified and discussed.
3. Again, I wonder whether an increased level of serum low-density lipoprotein L5 was specific for patients with RA. Some discussion regarding this issue should be needed.
Author Response
Thanks for your excellent review and useful comments
Looking at Table 1, RA patients with subclinical atherosclerosis was significantly older than patients without subclinical atherosclerosis or healthy controls. I wonder age of patients or duration of RA might be attributable to the appearance of subclinical atherosclerosis. Alternatively, age of patients or duration of RA might be influence serum level of low-density lipoprotein L5. This issue should be clarified.
Ans: Thanks for your comment. It is true that RA patients with subclinical atherosclerosis were significantly older than patients without subclinical atherosclerosis or healthy controls. Although the result did not reach statistical significance, the duration of RA was longer in patients with subclinical atherosclerosis than in those without. These observations might suggest that age and disease duration contribute to the emergence of subclinical atherosclerosis.
According to your good suggestions, we first constructed an univariate logistic regression model to evaluate the contribution of traditional risk factors such as age, RA disease duration, plasma L5% and L5 levels, and RA-related inflammatory parameters to the presence of subclinical atherosclerosis detected by using ultrasonography. Factors including age, RA disease duration, and the statistically significant variables in the univariate analysis were then enrolled to the multivariate regression analysis, and the result show L5% and L5 levels to be significant predictors of subclinical atherosclerosis (as shown in the revised Table 3). We have added the related data in Table 3 and descriptions in the section of “Methods” and “Results”. [P9, Line 6-11, in “Methods”; P12, Line 19, in “Results”; and the revised Table 3 of the revised manuscript]
The authors observed a decrease of serum level of low-density lipoprotein L5 after anti-RA treatment in patients with RA. Detailed treatment procedure in these patients should be represented.
Ans: Thanks for your good comments. As your good suggestions, we have added the detailed treatment procedure as follows: “Six patients received tumor necrosis factor-α inhibitor (adalimumab) at a dose of 40 mg every other week, and 8 patients received interleukin-6 receptor inhibitor (tocilizumab) therapy at a dose of 4mg/kg once monthly during the first 3 months and then 8mg/kg once monthly afterward, with concomitant MTX at a stable dose of 7.5-15 mg weekly. The dosage of csDMARDs as well as oral corticosteroids remained unchanged throughout the period of investigation.” [P11, Line 8-13, in “Results” of the revised manuscript]
I wonder serum level of low-density lipoprotein L5 might reflect only chronic inflammation state, not some pathogenesis of RA. This issue should be clarified. Also, whether some association between serum level of low-density lipoprotein L5 and serum level of C-reactive protein, erythrocyte sedimentation rate, or DAS28 existed. This issue should be clarified and discussed.
We fully agree your expert opinion that serum L5 levels might reflect chronic inflammation status in RA. Therefore, we went on to examine the correlation between serum L5 levels and disease activity (DAS28) of RA patients and found a positive correlation between DAS28 and serum L5 proportion or L5 levels (r=0.361, p<0.01 and r=0.254, p<0.05; respectively). We have revised accordingly. In addition, we have added the related descriptions in the limitation of this study as follows: “given that this study was cross-sectional in design and an elevated L5 level might reflect chronic inflammation state, the exact causal association of L5 with accelerated atherosclerosis in RA patients warrants further investigation”. [P12, Line 14-15, in “Results”; P16, Line 11-13, in “Discussion” and the revised Table 2 of the revised manuscript]
Again, I wonder whether an increased level of serum low-density lipoprotein L5 was specific for patients with RA. Some discussion regarding this issue should be needed.
Ans: Thanks for your comment. Based on our study and previous reports, the increase of L5 level is not specific to RA patients. Following your good suggestion, we have revised the related description in the section of “Discussion” as follows: “These findings are in accordance with those previously showing that plasma L5 levels are significantly increased in uremia patients with considerable CVD risks….. Previously, L5 levels were similarly shown to be elevated in asymptomatic patients with type 2 diabetes mellitus, in whom CVD risk was equivalent to that in patients with RA. Therefore, the increase of L5 level is not specific to RA patients.” [P14, Line 19-22, and P15, Line 1-4 in “Discussion” of the revised manuscript]
Reviewer 2 Report
In the present manuscript the authors have investigated the role of electronegative L5 lipoprotein on increased risk of cardiovascular disease (CVD) in rheumatoid arthritis (RA) patients. To address this question the authors have determined the proportion of plasma L5 in total LDL-C (L5%) and L5 levels in RA patients and healthy subjects and between RA patients with and without subclinical atherosclerosis. In addition, they examined the association of plasma L5 levels with the extent of subclinical atherosclerosis and 10-year CVD risk and determined potential predictive factors and their optimal cut-off values for predicting subclinical atherosclerosis in RA patients.
The results clearly show that plasma levels of L5 were significantly higher in RA patients than in healthy controls and that plasma L5% and L5 levels were positively correlated with the extent of carotid artery atherosclerosis and the 10-year CVD risk scores in RA patients.
Major comments
1) it is not clear why the authors have utilized the QRISK-2 scores for estimating the cardiovascular risk. The analysis should be performed by using also other scores, such as Framingham and SCORE (Systemic Coronary Risk Estimation).
2) as recognized by the authors, the study is strongly limited by the small size of patients recruited.
Author Response
Thanks for your excellent review and useful comments
It is not clear why the authors have utilized the QRISK-2 scores for estimating the cardiovascular risk. The analysis should be performed by using also other scores, such as Framingham and SCORE (Systemic Coronary Risk Estimation).
Ans: Thanks for your good comment. We estimate the 10-year CVD risk by using a QRISK-2 score rather than Framingham score because the QRISK-2 calculator additionally enrolls rheumatoid arthritis (RA), ethnicity, chronic kidney disease, and atrial fibrillation as factors in the risk assessment. In addition, the current National Institute for Health and Care Excellence (NICE) guidelines recommend using QRISK score.
According to your good suggestion, we have added another risk score (Framingham risk score) for analysis and presented the results in the revised Table 1 and Table 2. In addition, we have added the related descriptions in the “Discussion” as follows “We used Framingham risk score and QRISK-2 score to estimate the 10-year CVD risk and found that both of them were significantly higher in RA patients with subclinical atherosclerosis than in those without. The QRISK-2 scores were also positively correlated with plasma L5% or L5 levels while Framingham score were not. This discrepancy may be explained by the more inclusive calculation design of QRISK-2 scoring, additionally enrolling RA disease, ethnicity, chronic kidney disease, and atrial fibrillation as factors in the risk assessment.” [P6, Line 2-3 in “Methods”; P10, Line 10-13 in “Results”; P15, Line 17-22 in “Discussion”; and the revised Table 1 and Table 2 of the revised manuscript]
As recognized by the authors, the study is strongly limited by the small size of patients recruited.
Ans: As your expert comment, the sample size of the patients in whom we could observe the emergence of cardiovascular disease in this study was truly not large enough. Therefore, a long-term study of a larger group of RA patients is required. We have addressed this issue in the limitation in the revised manuscript. [P16, Line 13-14 and Line 20-21 in “Discussion” of the revised manuscript]